# Persistent Immunity against SARS-CoV-2 in Individuals with Oncohematological Diseases Who Underwent Autologous or Allogeneic Stem Cell Transplantation after Vaccination

**DOI:** 10.3390/cancers15082344

**Published:** 2023-04-18

**Authors:** Sara Rodríguez-Mora, Lucía Pérez-Lamas, Miriam Solera Sainero, Montserrat Torres, Clara Sánchez-Menéndez, Magdalena Corona, Elena Mateos, Guiomar Casado-Fernández, José Alcamí, Javier García-Pérez, Mayte Pérez-Olmeda, María Aranzazú Murciano-Antón, Javier López-Jiménez, Valentín García-Gutiérrez, Mayte Coiras

**Affiliations:** 1Immunopathology Unit, National Center of Microbiology, Instituto de Salud Carlos III, 28220 Madrid, Spain; 2Biomedical Research Center Network in Infectious Diseases (CIBERINFEC), Instituto de Salud Carlos III, 28029 Madrid, Spain; 3Hematology and Hemotherapy Service, Instituto Ramón y Cajal de Investigación Sanitaria (IRYCIS), Hospital Universitario Ramón y Cajal, 28034 Madrid, Spain; 4Faculty of Sciences, Universidad de Alcalá, 28801 Madrid, Spain; 5AIDS Immunopathology Unit, National Center of Microbiology, Instituto de Salud Carlos III, 28220 Madrid, Spain; 6Serology Service, Instituto de Salud Carlos III, 28029 Madrid, Spain; 7Family Medicine, Centro de Salud Doctor Pedro Laín Entralgo, 28924 Alcorcón, Spain

**Keywords:** COVID-19 vaccine, autologous transplantation, allogeneic transplantation, cytotoxic response, humoral response

## Abstract

**Simple Summary:**

Individuals with hematological diseases are particularly susceptible to COVID-19, and their vaccination has been a priority since the beginning of the pandemic. However, due to the disease itself or the treatment they receive, the protection achieved after vaccination may not be protected from severe forms of COVID-19. In this study, we analyzed the characteristics of the immune response developed by individuals with hematological cancer who received a stem cell transplant to overcome their disease after receiving the full vaccination schedule. All participants showed reduced levels of antibodies in plasma, but the cellular response of individuals with an autologous transplant was very similar to healthy donors, while it was severely impaired in individuals who received the transplant from other individuals (allogeneic). Interestingly, the transplant did not affect the immune response against SARS-CoV-2, and most infections reported after the transplant were mild, proving that some level of protection was preserved after the transplant.

**Abstract:**

The high morbimortality due to SARS-CoV-2 infection in oncohematological diseases (OHD) and hematopoietic stem cell transplant (HSCT) recipients in the pre-vaccine era has made vaccination a priority in this group. After HSCT, the immune responses against common vaccines such as tetanus, varicella, rubella, and polio may be lost. However, the loss of immunity developed by COVID-19 vaccination after HSCT has not been completely defined. In this study, both humoral and cellular immunity against SARS-CoV-2 were analyzed in 29 individuals with OHD who were vaccinated before receiving allogeneic (*n* = 11) or autologous (n = 18) HSCT. All participants had low but protective levels of neutralizing IgGs against SARS-CoV-2 after HSCT despite B-cell lymphopenia and immaturity. Although antibody-dependent cellular cytotoxicity was impaired, direct cellular cytotoxicity was similar to healthy donors in participants with autologous-HSCT, in contrast to individuals with allogeneic–HSCT, which severely deteriorated. No significant changes were observed in the immune response before and after HSCT. During follow-up, all reported post-HSCT SARS-CoV-2 infections were mild. This data emphasizes that COVID-19 vaccination is effective, necessary, and safe for individuals with OHD and also supports the persistence of some degree of immune protection after HSCT, at least in the short term, when patients cannot yet be revaccinated.

## 1. Introduction

More than three years into the COVID-19 pandemic caused by the severe acute respiratory syndrome coronavirus 2 (SARS-CoV-2), individuals with oncohematological diseases (OHD) and hematopoietic stem cell transplant (HSCT) recipients are known to be part of a particularly vulnerable population. Several studies show that these individuals have higher rates of hospitalization related to SARS-CoV-2 infection, intensive care admission, invasive ventilation, and increased mortality in comparison with infected individuals without cancer [1,2,3].

In the group of individuals receiving HSCT, registries from the European Society for Blood and Marrow Transplantation (EBMT) and the Center for International Blood and Marrow Transplant Research (CIBMTR) show mortality rates between 20–30% in relation to SARS-CoV-2 infection [2,4]. European guidelines recommend considering vaccination against COVID-19 as a priority over other vaccines included in the vaccination schedule after transplantation, starting vaccination against this virus between 3–6 months after the progenitor transplantation [5,6].

It is well known that the immaturity of donor-derived lymphocytes, the decrease in recipient plasma cells, and the low levels of antibodies resulting from transplantation lead to a loss of immunity to pathogens for which immunity had been acquired during childhood, either through vaccination or infection [7]. This loss of protection following HSCT has been described for multiple pathogens, such as *Streptococcus pneumoniae* [8,9,10,11], rubella and mumps [11], hepatitis B [12], tetanus [8], diphtheria, *Haemophilus influenzae* or polio [13,14]. 

Most of the available studies highlighting the loss of vaccine immunity after HSCT only assessed antibody titers as a measure of vaccine response [15], without considering the possible protection conferred by the development of efficient cellular immunity. This would be especially relevant in those individuals with significant B-cell lymphopenia and subsequent hypogammaglobulinemia, caused by the OHD or its treatment. Notwithstanding this, the available information leads to a consensus on the need for these patients to be revaccinated [6,15,16].

Although there are no specific studies that determine the loss of immunity after HSCT in individuals who were previously vaccinated against SARS-CoV-2, based on available studies demonstrating impaired immunity against other vaccines after transplantation, guidelines currently recommend considering post-HSCT individuals as “never vaccinated” against SARS-CoV-2, regardless of their pre-HSCT or donor vaccination history [15,16]. However, the evaluation of the protection against SARS-CoV-2 in the first months post-HSCT in previously vaccinated patients may be relevant for two main reasons: first, to provide evidence to establish firm recommendations about the need to repeat or not the complete vaccination regimen against SARS-CoV-2 versus other strategies, and to determine the best timing of revaccination; and second, to evaluate the risk of SARS-CoV-2 infection during this early post-transplant period, as it has been observed that the development of SARS-CoV-2 infection in the first 12 months post-HSCT is an independent factor for mortality [2]. The timing of revaccination is a delicate balance between immune recovery and risk of infection [6]. Likewise, although the evidence is still scarce in this group of individuals, several studies indicate that the vaccine response to SARS-CoV-2 in the first 6 months after allogeneic HSCT (Allo-HSCT) may be practically nonexistent [17,18,19,20,21,22], and therefore, the preservation of previous protection may represent an important barrier to infection.

Consequently, in this study, we evaluated the persistence of immunity against SARS-CoV-2 in previously vaccinated individuals with OHD who were then subjected to allogeneic or autologous HSCT (Auto-HSCT), analyzing both humoral and cellular responses.

## 2. Materials and Methods

### 2.1. Study Population

This is an observational, prospective study performed in a single center (Hospital Universitario Ramón y Cajal, Madrid, Spain) that included 29 individuals diagnosed with OHD who were subjected to HSCT after having received the complete vaccination schedule against SARS-CoV-2. These individuals were recruited between April and June 2021, according to the following inclusion criteria: to be over 18 years old and to have been vaccinated with 1–3 doses of one of the EMA/FDA approved vaccines (Comirnaty, BioNTech/Pfizer; Spikevax, Moderna; Vaxzevria, AstraZeneca) before being subjected to HSCT for the treatment of their OHD between June 2021 and January 2022. Whole blood samples were obtained from the participants before starting the conditioning and 2.5 months after transplantation.

Eighteen individuals without OHD who were vaccinated against SARS-CoV-2 infection were recruited as healthy donors. They presented similar characteristics to HSCT individuals in terms of age, gender, and time from vaccination to sampling.

### 2.2. Ethical Statement

This study was approved by the Ethics Committee of the Hospital Universitario Ramon y Cajal (favorable report number 053-21) and the Primary Care Assistance Management of Comunidad de Madrid (Spain) (favorable report number 20210008; CCI 20210017). All individuals gave informed written consent to participate in the study in accordance with the Helsinki Declaration. Confidentiality and anonymity were protected by current Spanish and European Data Protection Acts.

### 2.3. Samples Processing and Materials

Blood samples were directly processed after collection by centrifugation through the Ficoll–Hypaque gradient (Pharmacia Corporation, North Peapack, NJ, USA). Peripheral blood lymphocytes (PBMCs) and plasmas were separated and cryopreserved until the examination. Raji cell line (ATCC CCL-86) was obtained from the collection of Instituto de Salud Carlos III (Madrid, Spain). Vero E6 cell line (ECACC 85020206) (African green monkey kidney) was generously provided by Dr. Antonio Alcami (CBM Severo Ochoa, Madrid, Spain). HEK-293T cell line was provided by the National Institute for Biological Standards and Control (NIBSC). Raji cells were cultured using RPMI 1640 supplemented with 10% fetal calf serum (FCS), 100 units/mL penicillin/streptomycin, and 2 mM L-glutamine (Lonza, Basel, Switzerland). Adherent cells were cultured in DMEM equally supplemented.

### 2.4. SARS-CoV-2 Serology

Euroimmun-Anti-SARS-CoV-2 ELISA kit (Euroimmun, Lübeck, Germany) was used to measure plasma IgG titers against the S1 domain of the SARS-CoV-2 Spike glycoprotein. Positive results were those values of IgG titers >1.1; titers between 0.8 and 1.1 were undetermined and not included in the final analysis; titers under 0.8 were assumed as negative.

### 2.5. Pseudotyped SARS-CoV-2 Neutralization Assays

Pseudotyped SARS-CoV-2 virus pNL4-3Δenv_SARS-CoV-2-SΔ19(G614)_Ren was used to assessing the neutralizing capacity of the IgGs detected in the plasma of the participants [23]. This single-cycle virus expressed the HIV type 1 genome and the Renilla luciferase gene as a reporter. In order to evaluate the neutralization activity, pseudotyped SARS-CoV-2 (10 ng p24 Gag per well) was pre-incubated for 1 h at 37 °C with 4-fold serial dilutions (1/32 to 1/8192) of decomplemented IgG-positive plasma and then co-cultured for 48 h with a monolayer of Vero E6 cells. Viral infectivity was assessed in lysed Vero E6 cells by using Renilla Luciferase Assay kit (Promega, Madison, WI, USA) using luminometer Centro XS3 LB 960 and MikroWin 2010 software version 5.25 (Berthold Technologies, Baden-Württemberg, Germany). Titers of neutralizing antibodies were represented as 50% inhibitory dose (ID50) using non-linear regression in GraphPad Prism Software version 9 (GraphPad, Inc., San Diego, CA, USA).

### 2.6. Phenotyping of B Cells

In order to evaluate the B-cell subpopulations, PBMCs were labeled with the following antibodies: CD3, CD10, CD19, CD20, CD21, and CD27. Using BD LSRFortessa X-20 flow cytometer, the subpopulations of B cells (CD3-CD19+) were differentiated according to the expression of the following markers: plasmablasts (CD27^++^CD20^−^CD21^low^), activated memory cells (CD10^−^CD27^+^CD21^low^), resting memory cells (CD10^−^CD27^+^CD21^high^), tissue-type memory cells (CD10^−^CD27^−^CD21^low^), naïve B cells (CD10^−^CD27^−^CD21^high^), and immature or transitional cells (CD10^+^ CD27^−^). Analyses were performed using BD LSRFortessa X-20 flow cytometer and FlowJo software version 10.7.1 (Tree Star Inc., Ashland, OR, USA).

### 2.7. Antibody-Dependent Cellular Cytotoxicity Assay

Antibody-dependent cellular cytotoxicity (ADCC) was measured in PBMCs from the participants, as described before [23]. Briefly, Raji cells were stained with PKH67 Green Fluorescent Cell Linker (Merck KGaA, Darmstadt, Germany) and then coated with rituximab (50 μg/mL) (Selleckhem, Houston, TX, USA). Rituximab-coated Raji cells were co-cultured with PBMCs (1:2 ratio) for 18 h. Early apoptosis induced by PBMCs in Raji cells was determined by staining with Annexin V conjugated with phycoerythrin (PE) (Immunostep, Salamanca, Spain). Analyses were performed with BD LSRFortessa X-20 flow cytometer and FlowJo software version 10.7.1 (Tree Star Inc.).

### 2.8. Direct Cellular Cytotoxicity Assay

Vero E6 cells were infected with equal amounts (100 ng p24 Gag/well) of the most important variants of SARS-CoV-2 within clade 19B that were circulating in Spain at the time of the study: D614 and G614 [24]. Both pseudotyped viruses pNL4-3Δenv_SARS-CoV-2-SΔ19(G614)_Ren and pNL4-3Δenv_SARS-CoV-2-SΔ19(D614)_Ren were incubated with Vero E6 cells for 48 h. Then, Vero cells were co-cultured for 1 h with PBMCs isolated from the participants (ratio 1:10). Vero cells were detached with trypsin-EDTA solution (Sigma Aldrich-Merck, Darmstadt, Germany), and induction of cytotoxicity was measured using Caspase-Glo 3/7 Assay system (Promega) to evaluate the PBMCs-induced activation of caspace-3 in the monolayer. Viral infection in Vero E6 cells was also determined by measuring Renilla with Renilla Luciferase Assay kit (Promega) in Centro XS3 LB 960 luminometer (Berthold Technologies).

PBMCs were collected previous to detach the Vero E6 monolayer to evaluate the presence of the following cytotoxic cell populations: Natural Killer (NK) (CD3^−^CD56^+^CD16^±^), NKT-like (CD3^+^CD56^+^), and TCRγδ (CD3^−^CD8^±^TCRγδ^+^) cells by using specific conjugated antibodies: CD3-PE, CD8-APC H7, TCRγδ-FITC, CD56-BV605, and CD16-PercP (BD Biosciences). Analyses were performed using a BD LSRFortessa X-20 flow cytometer and FlowJo software version 10.7.1 (Tree Star Inc.).

### 2.9. Statistical Analysis

GraphPad Prism9.0 software (GraphPad Inc., San Diego, CA, USA) was used to perform the statistical analysis. Quantitative variables were represented as the mean and standard deviation of the mean (SEM). For the comparison between groups, statistical significance was determined by using one-way ANOVA with Tukey’s multiple comparisons test. Unpaired, a parametric *t*-test was used to compare longitudinal blood samples. Those comparisons with *p* values (*p*) < 0.05 were considered statistically significant.

## 3. Results

### 3.1. Patients’ Cohorts

The main sociodemographic and clinical data of the participants are summarized in Table 1 and detailed in Appendix A. Most of these individuals (69%) were male, and the median age was 59 years old (IQR 50–64). Most participants (93%) received a complete two-dose vaccination schedule with one authorized vaccine against COVID-19, while two participants received a booster dose (7%). Most participants (65%) received Spikevax, 17% received Comirnaty, and 17% received Vaxzevria. The mean time from vaccination to transplantation was 119 days (IQR 80–152). Six participants (21%) had SARS-CoV-2 infection confirmed by RT-qPCR prior to vaccination. The first blood sample was collected 6 days (IQR 1–7) before transplantation (pre-transplant sample), which was 106 days (IQR 69–147) after receiving the full vaccination schedule. A second blood sample was collected 76 days (IQR 66–83) after transplantation (post-transplant sample), when a median of 192 days (IQR 153–238) had passed since the complete vaccination schedule. All participants were followed closely during and after transplantation, and clinical information was collected regarding the development of Graft-versus-host disease (GvHD), the need for additional immunosuppressive treatment, the development of SARS-CoV-2 breakthrough infections after transplantation and their severity, and vaccination against COVID-19 after HSCT.

Eighteen participants were subjected to Auto-HSCT, and 11 participants were subjected to Allo-HSCT. The median age of individuals with Auto- and Allo-HSCT was 57 years old (IQR 46–65) and 60 years old (IQR 52–63), respectively. Acute lymphocytic leukemia (ALL) (54%) and myelodysplastic syndromes (MDS) (27%) were the most common indications for Allo-HSCT, while the most frequent pathologies for Auto-HSCT were multiple myeloma (MM) (55%), Non-Hodgkin lymphoma (NHL) (39%), and Hodgkin lymphoma (HL) (6%). All donors for Allo-HSCT had previously received a complete vaccination schedule against COVID-19, mostly with Comirnaty (64%). Vaccination of donors occurred within a median of 46 days (IQR 10–216) before transplantation.

Eighteen participants (62%) received chemotherapy prior to transplantation, and 13 (45%) received other therapies such as immunotherapy, targeted therapies, or proteosome inhibitors. Seven individuals (24%) received rituximab within 6 months prior to transplantation, while nine individuals (31%) received a reduced-intensity regimen. Among the participants with Allo-HSCT, 27% were HLA-identical transplants, while the rest were from alternative donors (64% haploidentical, 9% non-related donor). At the time of post-transplant sample collection, all individuals with Allo-HSCT were on systemic immunosuppressive therapy (100% cyclosporine/tacrolimus, 36% mycophenolate, and 27% corticosteroids). Two individuals (18%) developed active GvHD, and CMV was reactivated in 54% of individuals.

Eighteen healthy donors were recruited as the control population. Most of them (72%) were male, and the median age was 50 years old (IQR 42–62). The main sociodemographic and clinical data of healthy donors are summarized in Table 2 and detailed in Appendix A. The most common comorbidities were dyslipidemia, obesity, diabetes, asthma, and high blood pressure, but none were receiving immunosuppressive treatment at the time of sampling. In terms of vaccination, they received a full vaccination schedule with Comirnaty (89%) and Spikevax (11%). The median time from the complete vaccination schedule to the first sample collection was 28 days (IQR 23–30); median time from the first to the second sample was 63 days (IQR 60–63) when a median of 91 days (IQR 86–93) had passed since complete vaccination schedule.

### 3.2. SARS-CoV-2 Breakthrough Infection

All participants followed an average of 8 months after HSCT to evaluate the development of SARS-CoV-2 breakthrough infections. They started revaccination against SARS-CoV-2 three months after HSCT, according to current recommendations, without relevant complications related to the vaccine. Revaccination occurred after the second sample was taken for this study. Eleven participants (11/29; 38%) had a COVID-19 infection, confirmed by qPCR, during the follow-up period. 10/11 (91%) participants presented a mild form of COVID-19 without requiring hospitalization or oxygen therapy. One patient (1/11; 9%) required hospitalization due to the development of immune neutropenia related to the infection, and low-dose oxygen therapy was eventually required. Of these participants who were infected after transplant, 5/11 (45%) had already received two doses of vaccine against SARS-CoV2 post-HSCT, 4/11 (36%) had received one dose, and 1/11 (9%) had not yet been revaccinated. The participant (1/11) who had not been vaccinated after HSCT did not develop a severe form of the infection nor require hospitalization.

Eight healthy donors (8/18; 44%) had SARS-CoV-2 breakthrough infection confirmed by qPCR 12 months (IQR 8–13) after the second vaccine dose. All of them had mild COVID-19, received symptomatic treatment, and did not require hospitalization.

### 3.3. Levels of IgGs against SARS-CoV-2 before and after HSCT

All individuals subjected to HSCT showed levels of IgGs against SARS-CoV-2 above the threshold of detection before transplant (Figure 1A). These levels were significantly reduced in comparison with healthy donors (−3.7-fold, *p* < 0.0001 in Allo-HSCT; −2.5-fold; *p* = 0.0001 in Auto-HSCT). After transplantation, both groups showed similar levels of IgGs to prior transplantation, 2.5 months before, while these levels were reduced 1.2-fold (*p* < 0.0001) in the plasma of healthy donors after 3 months. In comparison with healthy donors, the levels of IgGs were reduced 2.0- (*p* = 0.0121) and 2.1-fold (*p* = 0.0077) in the plasma of participants with Allo- and Auto-HSCT, respectively.

No significant differences were observed between groups in the neutralizing capacity of IgGs against SARS-CoV-2 (Figure 1B).

### 3.4. Changes in B Cell Subpopulations after HSCT

Before HSCT, total B cell count (CD19^+^) was reduced 4.6- (*p* = 0.0009) and 3.0-fold (*p* = 0.0015) in PBMCs from individuals who underwent Allo- and Auto-HSCT, respectively, in comparison with healthy donors (Figure 2A). After HSCT, individuals with Allo- and Auto-HSCT showed similar levels of total B cells that were still significantly reduced in comparison with healthy donors (*p* < 0.0001 and *p* = 0.0008, respectively).

The levels of immature B cells after HSCT were significantly increased in comparison with healthy donors in both Allo- and Auto-HSCT (3.4-fold, *p* = 0.011; 3.5-fold, *p* = 0.002, respectively) (Figure 2B). Conversely, the levels of naïve B cells were reduced in both groups, in comparison with healthy donors (−5.1-fold; *p* < 0.0001 and −4.1-fold; *p* < 0.0001, respectively). Reduced levels of tissue-like memory (-3.0-fold; *p* = 0.037 for Allo-HSCT and −2.5-fold; *p* = 0.034 for Auto-HSCT), resting memory B cells (*p* = 0.002 for Allo-HSCT and *p* = 0.003 for Auto-HSCT), and activated memory (*p* = 0.019 for Allo-HSCT and *p* = 0.035 for Auto-HSCT) also showed statistical significance in comparison with healthy donors.

### 3.5. Cytotoxic Cellular Immune Responses against SARS-CoV-2 before and after HSCT

The antibody-dependent cellular cytotoxicity (ADCC) was reduced in both groups of participants before Allo- and Auto-HSCT, in comparison with healthy donors (−1.8-fold; *p* = 0.0014 and −1.7-fold; *p* = 0.0003, respectively) (Figure 3A). These differences were maintained after HSCT in both groups (−1.5-fold; *p* = 0.0126 and −1.6-fold; *p* = 0.0022, respectively). There were no differences between ADCC response before and after HSCT in either group.

Direct cytotoxic activity (DCC) was increased in PBMCs of individuals with Auto-HSCT, before and after transplant, in comparison with healthy donors and individuals with Allo-HSCT, who showed the lowest DCC (Figure 3B). Although these results did not achieve statistical significance, likely due to high data dispersion, decreased DCC in PBMCs from individuals with Allo-HSCT correlated with reduced capacity to eliminate SARS-CoV-2-infected cells before and after transplant, in comparison with healthy donors (−3.6-fold; *p* = 0.0170) (Figure 3C). PBMCs from individuals with Auto-HSCT showed similar efficiency in eliminating SARS-CoV-2-infected cells than PBMCs from healthy donors before and after transplant. As occurred with ADCC, there were no differences between DCC response before and after HSCT in either group of participants.

### 3.6. Characterization of Cytotoxic Cell Populations in PBMCs of Transplanted Individuals

Total levels of CD3+ T lymphocytes were similar to healthy donors in individuals previous to Auto-HSCT, but they were significantly reduced in individuals with Allo-HSCT (−1.6-fold; *p* = 0.0219) (Figure 4A). After the transplant, no significant differences were observed within groups.

Total levels of CD8+ T cells were only significantly increased in individuals with Auto-HSCT 2.5 months after transplant (1.3-fold; *p* = 0.0042) (Figure 4B). The expression of the degranulation marker CD107a decreased 1.4-fold (*p* = 0.0247) in CD8+ T cells isolated from healthy donors three months after the first sample (Figure 4C). The expression of this marker was also reduced 1.7-fold (*p* = 0.0038) in CD8+ T cells from individuals with Auto-HSCT before transplant, in comparison with healthy donors, but no significant changes were observed after transplantation in either group.

Individuals with Allo- and Auto-HSCT showed increased levels of CD8+ T cells with TCRγδ+ before and after transplant (3.2-fold; *p* = 0.0106 and 2.4-fold; *p* = 0.0496, respectively), in comparison with healthy donors (Figure 5A), while individuals with Auto-HSCT showed increased levels of CD3+CD8-TCRγδ+ cells before and after transplant (3.4-fold; *p* = 0.0035 and 5.9-fold; *p* = 0.0014, respectively), in comparison with healthy donors (Figure 5B).

Individuals with Auto-HSCT showed decreased levels of NK cells (CD3-CD56+) before transplant (−1.5-fold; *p* = 0.0267), in comparison with healthy donors (Figure 6A). The levels of NKT-like cells (CD3+CD56+) were significantly increased 1.3-fold (*p* = 0.0460) in individuals with Auto-HSCT after transplant (Figure 6B). No changes between groups were observed in the expression of CD107a in TCRγδ+, NK, or NKT-like cells (Appendix A).

## 4. Discussion

The fast development and implementation of COVID-19 vaccines are two of the most important achievements of the current medical research, and it will undoubtedly change the future of vaccine science [25]. Sixteen months after the authorization of the first vaccine against COVID-19 by the FDA, Comirnaty (Pfizer/BioNTech), and now that more than 13 billion vaccine doses have been administered globally, the cases of COVID-19 and SARS-CoV-2-related deaths have dramatically decreased [26]. However, the effectiveness of COVID-19 vaccination in individuals with OHD is still low due to the disease itself or the therapeutic procedures that are implemented, among them HSCT. Some studies show variable rates of immune responses in individuals with OHD who were vaccinated against COVID-19 after receiving HSCT [27,28]. Until now, only one study has evaluated the persistence of the immune response in individuals who were vaccinated before receiving HSCT, and it suggests that donor vaccination status may impact the post-HSCT humoral response to early vaccination against SARS-CoV-2 after Allo-HSCT [21].

In the present study, we analyzed the persistence of both humoral and cellular immune responses developed against SARS-CoV-2 in individuals with OHD who were subjected to Allo- or Auto-HSCT after receiving the complete vaccination schedule. Both groups of individuals showed detectable IgG titers against SARS-CoV-2 before HSCT that, although reduced in comparison with healthy donors, remained unchanged 2.5 months after transplantation. These reduced IgG levels are probably related to significant B-cell lymphopenia, with high levels of immature B cells observed in both groups of individuals, which may have contributed to low titers of neutralizing antibodies. Normal levels of total B cells are described as taking about 6–12 months to normalize after HSCT, and functional recovery of B lymphocytes takes several months to years [7]. However, these levels were also reduced in healthy donors within 6 months after receiving the last dose of the complete vaccination schedule, which is in accordance with previous reports [29,30], mostly in never infected, vaccinated individuals [31]. In addition, ADCC response was impaired in both groups of individuals in comparison with healthy donors before and after HSCT, which may also be related to low IgG titers. Therefore, humoral immunity in response to COVID-19 vaccination was impaired in individuals with OHD before HSCT, and transplantation did not significantly modify this reduced capacity, which was maintained at least 2.5 months after HSCT, independently of if it was allogeneic or autologous. Total IgG levels were slightly increased after transplantation in the Allo-HSCT group, which may be due to all donors had been previously vaccinated, but this difference did not achieve statistical significance, and no changes were observed in the neutralizing capacity or ADCC response in this group in comparison with individuals with Auto-HSCT.

On the other hand, we observed significant differences in the cellular immune response between groups. Individuals with Allo-HSCT showed significantly reduced antiviral activity against target cells infected with pseudotyped SARS-CoV-2, in comparison with individuals with Auto-HSCT who showed similar DCC activity than healthy donors, which was translated into an increased antiviral activity against the infected cells.

In the current guidelines, recommendations for Allo-HSCT and auto-HSCT recipients are uniform [6,15,16], while immune reconstitution and immunologic memory differ [6,32]. It is described as a faster increase of lymphocytes in the autologous group, with faster reconstitution in B-cells and higher CD3 and CD4+ cell counts in T lymphocytes [32], which may affect susceptibility to infections and response to vaccines. Despite this, due to the lack of data on post-Allo-HSCT patients, recommendations derived from studies in post-Allo-HSCT patients generally apply [6]. The differences in the cellular immune response showed in our study, where the DCC activity was similar to healthy donors, contribute to thinking that a more tailored approach might be indicated depending on the type of transplantation. In line with previous observations, individuals with Allo-HSCT showed the lowest levels of total lymphocyte counts. And although the levels of CD8+ T cells were similar between groups, without changes in the expression of the degranulation marker CD107a, only individuals with Auto-HSCT showed a significant increase of these cells after HSCT. Moreover, highly cytotoxic CD3+CD8-TCRγδ+ [33] were also increased in individuals with Auto-HSCT, before and after transplantation, which may contribute to a higher antiviral activity of PBMCs from these individuals, even with similar levels of CD107a and significantly reduced NK cell levels. Other factors probably contribute to the observed differences, such as the presence of immunosuppressive treatment in post-Allo-HSCT patients, underlying disease, pre-transplant therapy, type of conditioning regimen, etc. All these elements differ between autologous and allogeneic transplantation.

The efficacy of the different vaccines received by the participants in the study (Spikevax, Comirnaty, and Vaxzevria) has been described to be similar (approximately 90%) within the first 6 months after the full vaccination schedule of two doses [34]. Therefore, we do not expect a significant influence on the final results due to the type of vaccine. Due to age may be a factor that significantly affects the efficacy and quality of the immune response [35,36]; all individuals recruited for this study had a median age between 50 and 60 years old.

Despite the differences observed in the immune response against SAR-CoV-2 vaccines between individuals with HSCT and healthy donors, the rate of SARS-CoV-2 breakthrough infections was similar (38% versus 44%) and with low clinical relevance, supporting that both individuals with Allo- and Auto-HSCT maintained some degree of protection after HSCT. The strict protective measures that are applied during the clinical management of these individuals may also contribute to this protection due to their level of exposure to SARS-CoV-2 and other respiratory viruses being likely much lower than the control population.

## 5. Conclusions

In conclusion, this study demonstrated that individuals with OHD had low but probable protective levels of IgGs against SARS-CoV-2 after HSCT that showed some neutralizing effect. Although ADCC activity was impaired, the direct cellular immune response was similar to healthy donors in participants with Auto-HSCT, likely due to the different immune reconstitution and the absence of immunosuppressive therapy after transplantation in this group. No significant improvement in the immune response was observed in individuals who received Allo-HSCT from previously vaccinated donors. Interestingly, no significant changes were observed in ADCC or DCC before and after HSCT in any group, proving that the immune response may persist after HSCT. In accordance, although the rate of breakthrough infections recorded in our series was high, most individuals presented a mild course of infection with a very low rate of hospitalization, which supported the maintenance of some degree of immunity after HSCT, despite significant B-cell lymphopenia and immaturity. These results emphasize that COVID-19 vaccination is effective, necessary, and safe for individuals with OHD and also provides some reassurance of immune protection in the first three months post-transplantation when the patients cannot be vaccinated. Further studies on vaccine response are needed to establish if a booster dose may be sufficient or if these individuals would benefit more from repeating the full vaccination schedule.

## Figures and Tables

**Figure 1 cancers-15-02344-f001:**
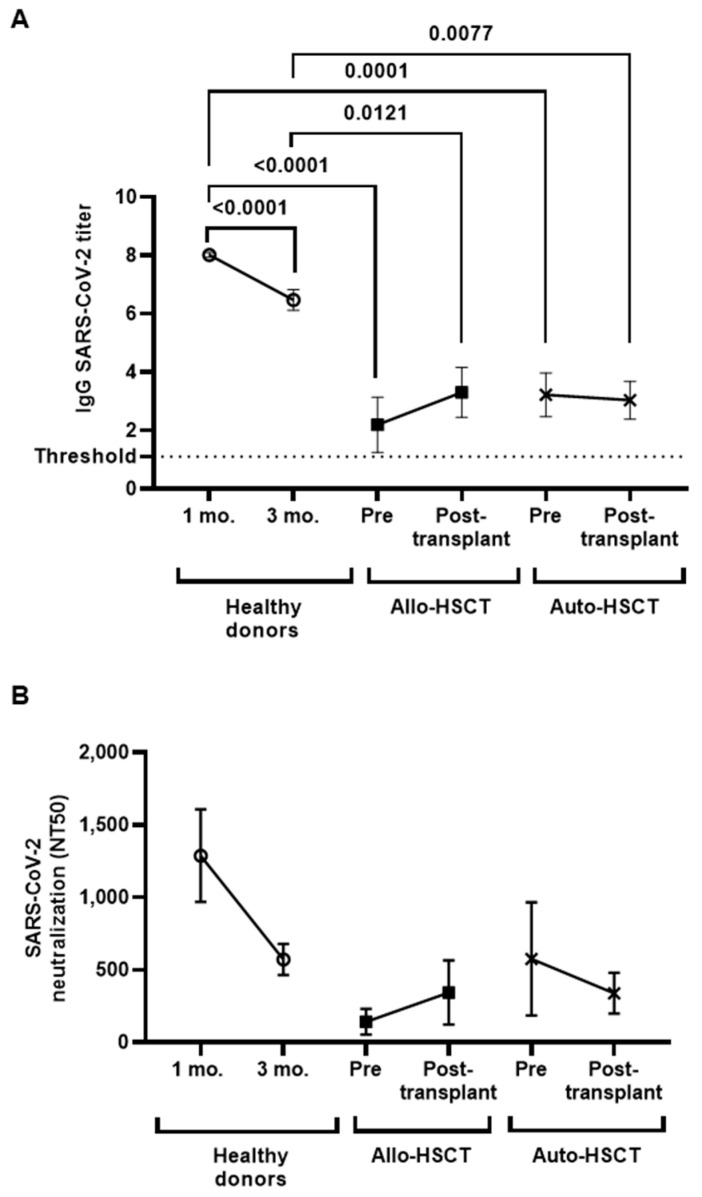
Serological response against SARS-CoV-2 vaccine in plasma from individuals with OHD subjected to Allo- and Auto-HSCT. (**A**) Plasma IgGs titers in individuals with Allo- or Auto-HSCT were compared between them and with healthy donors. (**B**) IgG neutralizing activity against SARS-CoV-2. Each dot corresponds to the mean ± standard error of the mean (SEM). Different cohorts are marked with the following symbols: Healthy donors (white circles), Allo-HSCT (black squares), and Auto-HSCT (cross). A one-way ANOVA test was applied to calculate the statistical significance between groups, and *t*-test was applied to calculate the statistical significance within groups.

**Figure 2 cancers-15-02344-f002:**
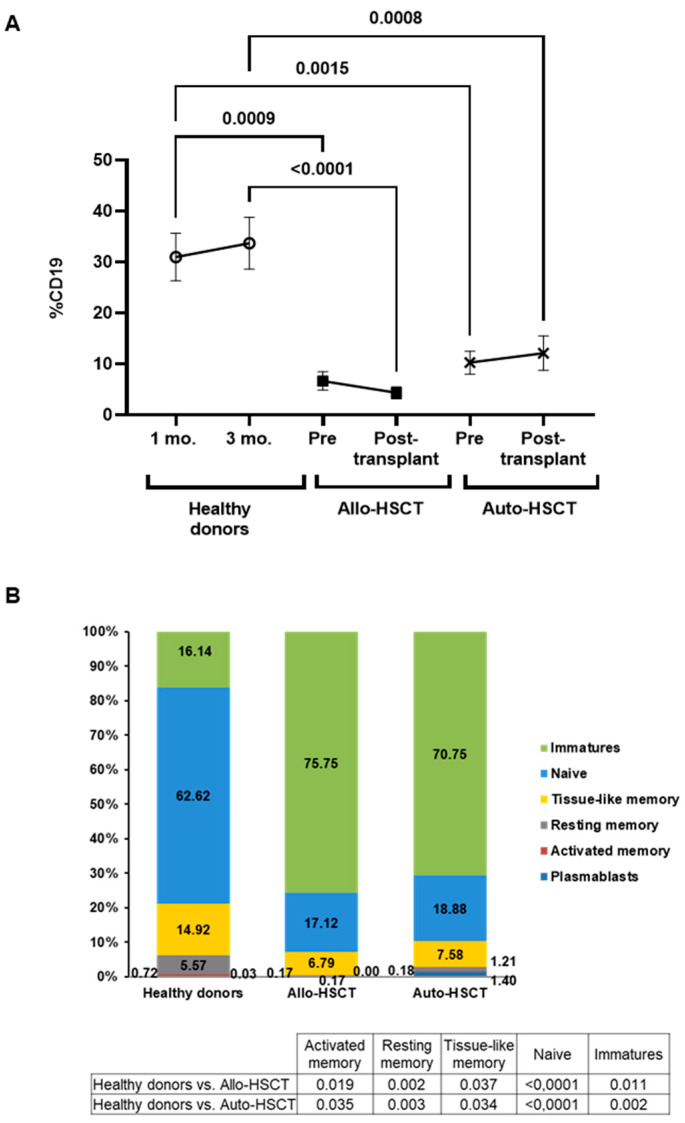
Total levels and subpopulations of B cells in PBMCs of individuals with OHD before and after receiving HSCT. (**A**) Total B cell levels were determined in PBMCs isolated from individuals with OHD with Allo- or Auto-HSCT before and after receiving transplantation and compared with healthy donors. Each dot in the graphs corresponds to mean ± SEM. Different cohorts are marked with the following symbols: Healthy donors (white circles), Allo-HSCT (black squares), and Auto-HSCT (cross). (**B**) Analysis of B cells subpopulations after HSCT in both cohorts and controls. Mean data are represented in bar graphs. A one-way ANOVA test was applied to calculate the statistical significance between groups, and *t*-test was applied to calculate the statistical significance within groups.

**Figure 3 cancers-15-02344-f003:**
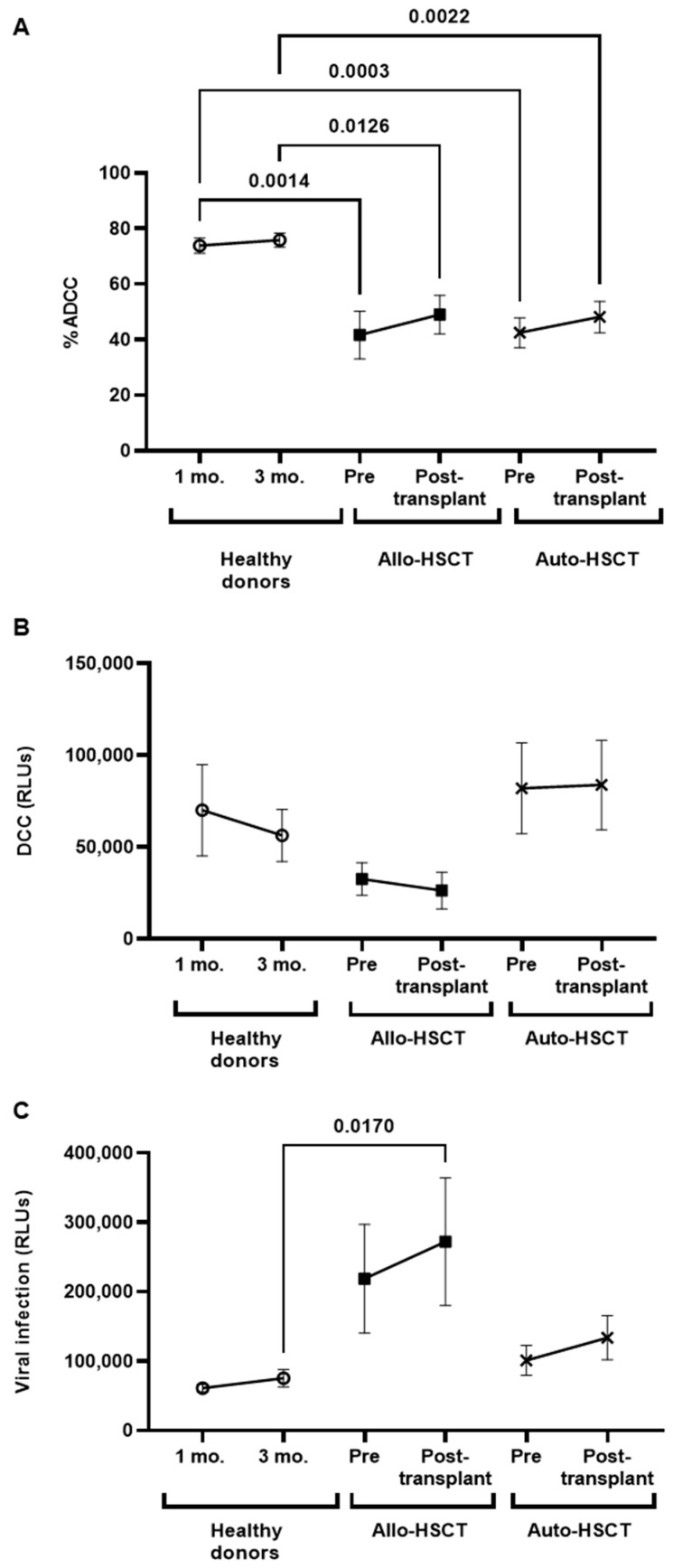
ADCC and DCC responses against SARS-CoV-2 of PBMCs from individuals with OHD subjected to HSCT. (**A**) The expression of phosphatidylserine was determined on the surface of rituximab-coated Raji cells by previous staining with Annexin V-PE of PBMCs from individuals with Allo- or Auto-HSCT before and after receiving transplantation and also compared with healthy donors. (**B**) DCC was determined in SARS-CoV-2-infected Vero E6 cells after co-culture with PBMCs isolated from the participants by determining caspase-3 activity in the monolayer. (**C**) PBMCs’ antiviral activity against SARS-CoV-2-infected Vero E6 cells was evaluated by determining the synthesis of Renilla (RLUs) after co-culture for 1 h. Each dot corresponds to the mean ± standard error of the mean (SEM). Different cohorts are marked with the following symbols: Healthy donors (white circles), Allo-HSCT (black squares), and Auto-HSCT (cross). A one-way ANOVA test was applied to calculate the statistical significance between groups, and *t*-test was applied to calculate the statistical significance within groups.

**Figure 4 cancers-15-02344-f004:**
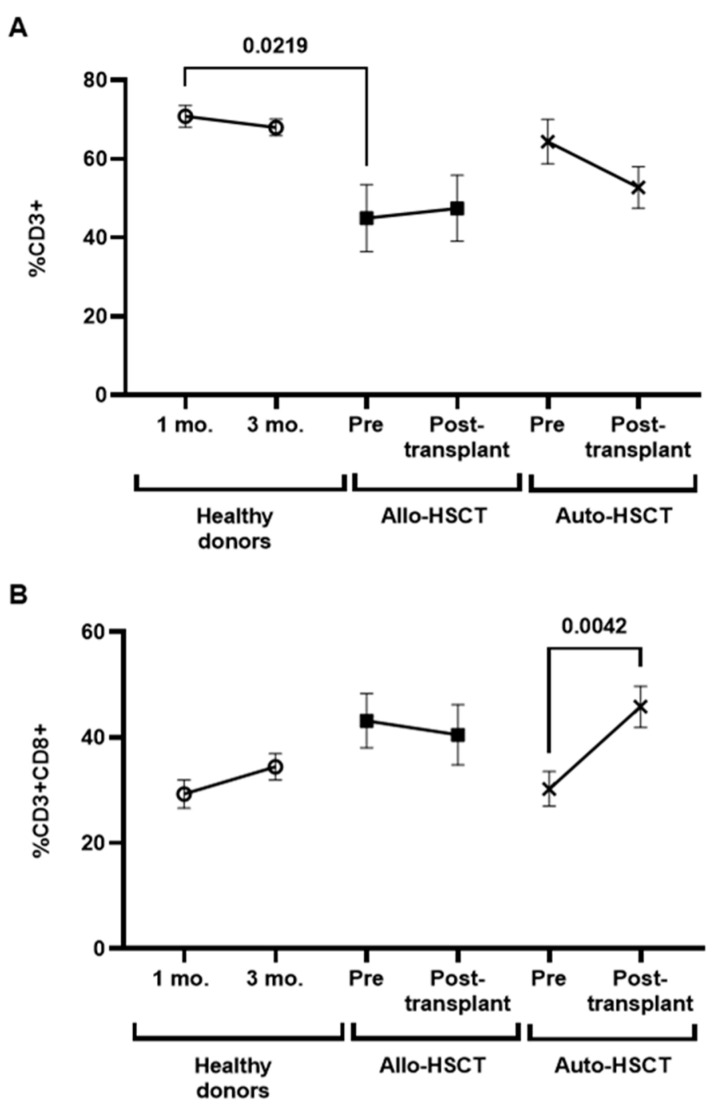
Characterization of total lymphocytes and CD8+ T cells in PBMCs from individuals with OHD before and after receiving Allo- or Auto-HSCT. (**A**) Total CD3+ T cells in PBMCs from individuals with Allo- or Auto-HSCT were evaluated by flow cytometry before and after HSCT. CD8+ cell count (**B**) and expression of CD107a in these cells (**C**) were determined in PBMCs from individuals with OHD, before and after HSCT, in comparison with healthy donors. Each dot in the graphs corresponds to mean ± SEM. Different cohorts are marked with the following symbols: Healthy donors (white circles), Allo-HSCT (black squares), and Auto-HSCT (cross). Statistical significance between groups was calculated using a one-way ANOVA test, and statistical significance within groups was calculated using a *t*-test.

**Figure 5 cancers-15-02344-f005:**
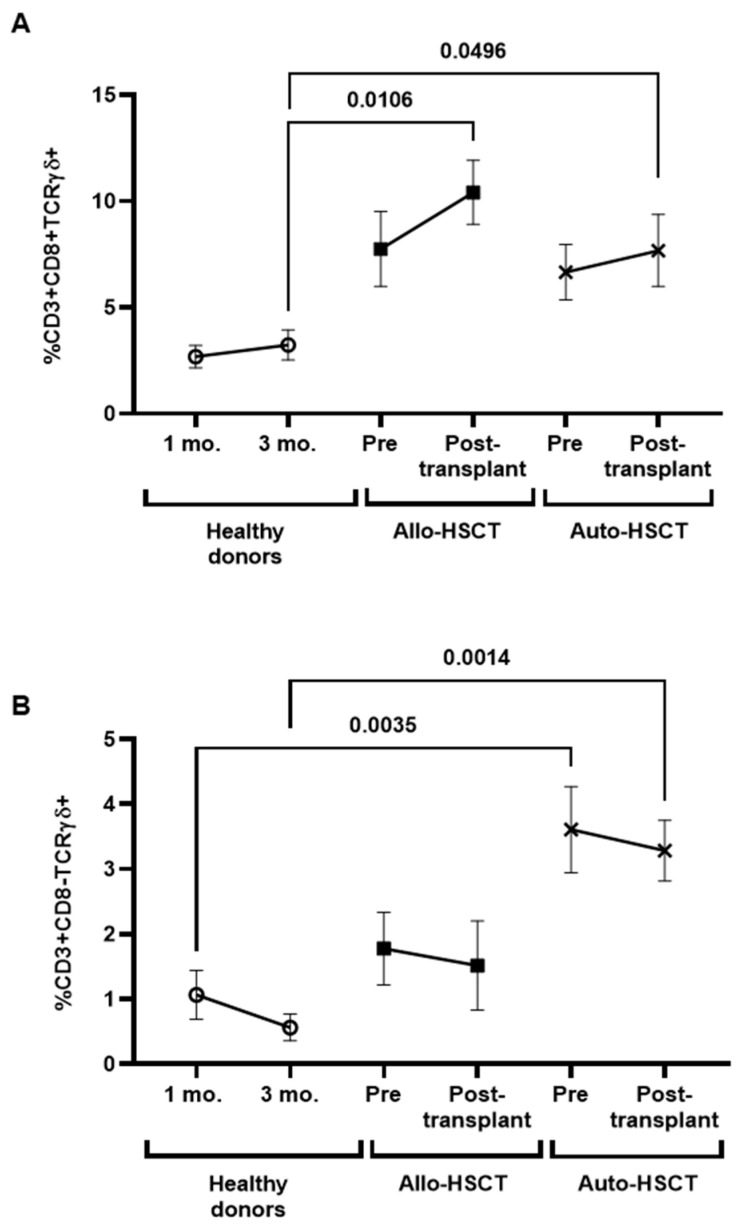
Characterization of TCRγδ+ cells in PBMCs from individuals with OHD before and after receiving Allo- or Auto-HSCT. Levels of CD3+CD8+TCRγδ+ (**A**) and CD3+CD8-TCRγδ+ (**B**) cells in PBMCs from individuals with Allo- or Auto-HSCT were evaluated by flow cytometry in comparison with healthy donors. Each dot in the graphs corresponds to mean ± SEM. Different cohorts are marked with the following symbols: Healthy donors (white circles), Allo-HSCT (black squares), and Auto-HSCT (cross). Statistical significance between groups was calculated using a one-way ANOVA test, and statistical significance within groups was calculated using a *t*-test.

**Figure 6 cancers-15-02344-f006:**
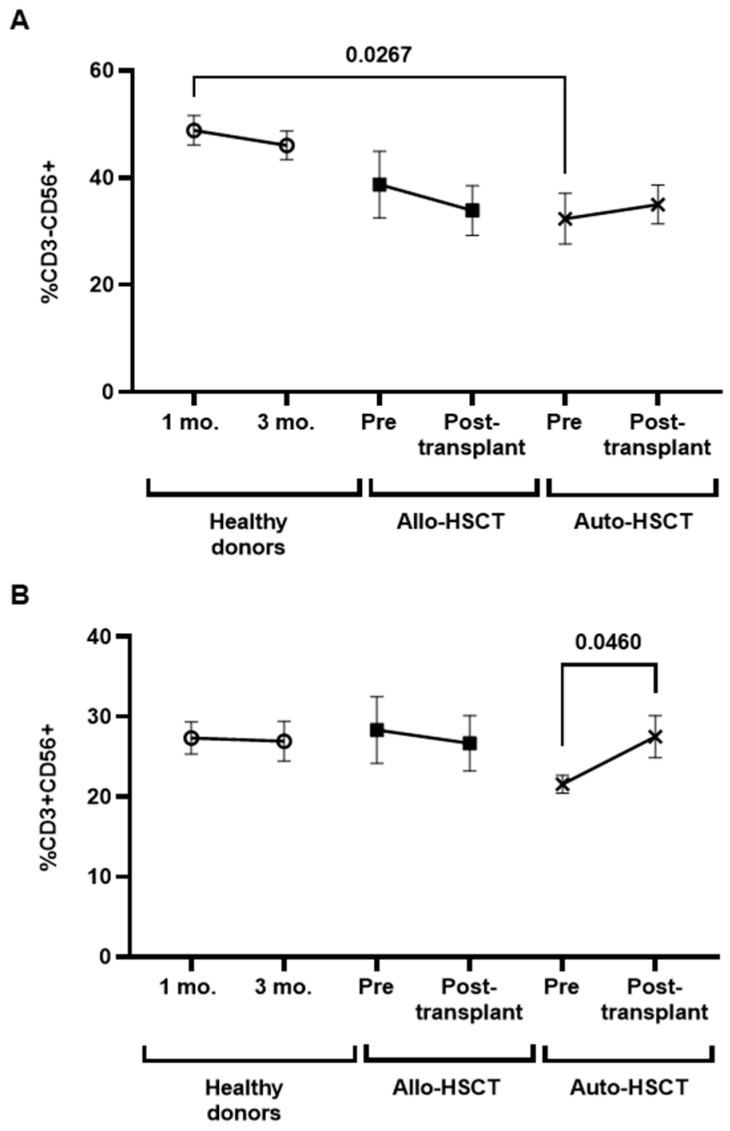
Characterization of NK and NKT-like cells in PBMCs from individuals with OHD before and after receiving Allo- or Auto-HSCT. Levels of NK (CD3-CD56+) (**A**) and NKT-like (CD3+CD56+) (**B**) cells in PBMCs from individuals with Allo- or Auto-HSCT were evaluated by flow cytometry in comparison with healthy donors. Each dot in the graphs corresponds to mean ± SEM. Different cohorts are marked with the following symbols: Healthy donors (white circles), Allo-HSCT (black squares), and Auto-HSCT (cross). Statistical significance between groups was calculated using a one-way ANOVA test, and statistical significance within groups was calculated using a *t*-test.

**Table 1 cancers-15-02344-t001:** Sociodemographic and clinical data of the individuals with allogeneic or autologous transplantation who participated in this study.

	Allo-HSCT (*n* = 11)		Auto-HSCT (*n* = 18)	
Median age at data collection, median years (IQR)	60 (52–63)		57 (46–65)	
Gender, *n* (%)				
Male	7 (64)		13 (72)	
Female	4 (36)		5 (28)	
Underlying oncohematological disease, *n* (%)				
	MDS	3 (27)	MM	10 (55)
	ALL	6 (54)	NHL	7 (39)
	Others	2 (18)	HL	1 (6)
**Pre-transplant parameters at the time of first sample collection**				
Pre-transplant treatment with immunosuppressive potential				
Chemotherapy, *n* (%)	9 (82)		9 (50)	
Targeted therapies, *n* (%)	7 (64)		6 (33)	
Type of transplant according to the donor			NA	
Related donor, *n* (%)	10 (90)			
Identical	3 (27)			
Haploidentical	7 (64)			
Non-related donor, *n* (%)	1 (9)			
Identical	0			
Mismatch	1 (9)			
Conditioning, *n* (%)				
Myeloablative	8 (73)		12 (67)	
Reduced intensity	3 (27)		6 (33)	
GvHD prophylaxis, *n* (%)			NA	
Cy post + Csa + MMF	9 (82)			
Csa + Mtx	2 (18)			
Vaccine type, *n* (%)				
Vaxzevria (AstraZeneca)	1 (9)		4 (22)	
Comirnaty (Pfizer)	2 (18)		3 (17)	
Spikevax (Moderna)	8 (73)		11 (61)	
SARS-CoV-2 infection prior to transplant, *n* (%)	4 (36)		2 (11)	
Mean time from vaccination of recipient to transplantation, median days (IQR)	95 (21–298)		105 (38–205)	
Vaccinated donor, *n* (%)	11 (100)		NA	
Vaccine type of donors, *n* (%)			-	
Comirnaty (Pfizer)	7 (64)		-	
Spikevax (Moderna)	1 (9)		-	
Vaxzevria (AstraZeneca)	1 (9)		-	
Jcovden (Janssen)	1 (9)		-	
Unknown	1 (9)		-	
Mean time from vaccination of the donor to transplantation, median days (IQR)	46 (10–216)		NA	
Immunoglobulin deficiency, *n* (%)				
IgG	2 (18)		7 (39)	
IgM	2 (18)		13 (72)	
IgA	2 (18)		9 (50)	
**Post-transplant parameters at the time of second sample collection**				
Immunosuppressive medications after HSCT, *n* (%)			NA	
Csa/Tacrolimus	11 (100)			
MMF	4 (36)			
Corticosteroids	3 (27)			
GvHD, *n* (%)	2 (18)		NA	
CMV replication, *n* (%)	6 (54)		0	
Disease relapse, *n* (%)	1 (9)		0	
Admitted to ICU, *n* (%)	2 (18)		0	
SARS-CoV-2 breakthrough infection after HSCT confirmed by PCR, *n* (%)	4 (36)		7 (39)	
Severity of COVID-19, *n* (%)				
Mild	3 (75)		7 (100)	
Hospitalized	1 (25)		0	

ALL, Acute lymphocytic leukemia; Allo-HSCT, Allogeneic haematopoietic stem cell transplantation; Auto-HSCT, Autologous hematopoietic stem cell transplantation; CMV, Cytomegalovirus; Csa, Cyclosporine; GvHD, Graft-versus-host-disease; Cy, cyclophosphamide; HL, Hodgkin lymphoma; ICU, Intensive care unit; MDS, Myelodysplastic syndrome; MM, Multiple myeloma; MMF, Mycophenolate mofetil; Mtx, Methotrexate; NA, Not applicable; NHL, Non-Hodgkin lymphoma.

**Table 2 cancers-15-02344-t002:** Sociodemographic and clinical data of the healthy donors who participated in this study as controls.

	Healthy Donors (*n* = 18)
Median age at data collection, median years (IQR)	50 (42–62)
Gender, *n* (%)	
Male	13 (72)
Female	5 (28)
Underlying oncohematological disease, *n* (%)	0
Vaccine type, *n* (%)	
Comirnaty (Pfizer)	16 (89)
Spikevax (Moderna)	2 (11)
SARS-CoV-2 infection prior to vaccination, *n* (%)	0
Time from complete vaccination schedule to 1st sample, median days (IQR)	28 (23–30)
Time from complete vaccination schedule to 2nd sample, median days (IQR)	91 (86–93)
SARS-CoV-2 breakthrough infection after vaccination confirmed by PCR, *n* (%)	8 (44)
Severity of COVID-19, *n* (%)	
Mild	8 (100)
Hospitalized	0

## Data Availability

All data analyzed or generated during the study have been included in this report.

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
