# Peer review of "Persistent Immunity against SARS-CoV-2 in Individuals with Oncohematological Diseases Who Underwent Autologous or Allogeneic Stem Cell Transplantation after Vaccination"

_cancers, 2023, doi:10.3390/cancers15082344_

Round 1

Reviewer 1 Report

The Authors have reported data indicating the “cellular response of individuals with an autologous transplant was very similar to healthy donors, while it was severely impaired in individuals who received the transplant from other individuals (allogenic). Interestingly, the transplant did not affect the immune response against SARS-CoV-2 and most infections reported after transplant were mild, proving that some level of protection was preserved after transplant”. While this study shows important evidence, it is nevertheless too preliminary to justify any strong conclusions. However, the analysis is thorough and is nevertheless important information that should reach the public domain. Such studies can motivate further motivate more research in this. However, the following points need to be addressed by the authors before this work can be published.

Major Points

Is this sample size good enough to draw any conclusion?

Conclusion of the investigators needs to take it in consideration the affectivity of different vaccines. Also, since the allogenic HSCT were subjected to immunosuppressive drugs, there could be interference in Immune actions from them.

 It is suggested to the  investigators to explore other infections as well.

Minor points

Conclusion should be placed before discussion.

Reviewer 2 Report

The article is interesting but I recommend the background be improved. Statistical analyses need to be checked. Why the authors used to mean and not median? They used Anova one-way and Mann Whitney test which are designed for variables that do not respect normality. 

Reviewer 3 Report

The authors examined in patients and normal controls the immune response in terms of production of antibodies and a specific cellular response following vaccination against a Covid-19 infection. The patients had various malignant hematological diseases such as leukemia, myeloma, Hodgkin and Non-Hodgkin lymphoma and underwent autologous or allogeneic stem cell transplantation. The immune response was measured prior and after transplantation.

While the study tools and parameters were rather simple, the data are nevertheless of interest and of importance. The universal theme is that Covid-19 vaccination in such patients is effective, important and safe.

Specific Points of Criticism and Suggestions for Alterations:

(1)  Abstract and Conclusions:  Maybe the authors wish to add a strong take-home message at the end of these sections like "Our studies emphasize that Covid-19 vaccination is effective (and hence important) and also safe for patients with malignant hematological diseases" (or similar).

(2)  Line 103:  Do the authors mean "a unique center" (one of a kind) or "a single center"?

(3)  Paragraph 2.3:  Have the authenticity of the cell lines and their mycoplasma status been checked and how?

(4)  Line 130:  It should be written "RPMI 1640" medium as there are actually different RPMI-media with different numbers.

(5)  Line 152 (and elsewhere):  The CD antibodies could be arranged in ascending numerical order.

(6)  Line 189:  "For the comparison between groups"  (not "when compare between groups").

(7)  Lines 197-202 (and elsewhere:)  The use of decimals after the period (e.g. 68.9% - or 93.1.6%?) is exaggerated and given the low numbers of patients is only pseudo-exactness, suffice to use whole numbers like in Table 1. The same for the days (IQR 1.5-7.0, better IQR 1-7).

(8)  Table 1:  "myeloablative" (not "mieloablative").

(9)  Tables 1 and 2:  The n(%) has been forgotten at various places:

Table 1, categories:  Vaccine type / SARS-CoV-2 infection prior to transplant / GvHD

Table 2, categories:  Vaccine type / SARC-Co-2 breakthrough.

(10)  Table 1:  The abbreviation "Cy" should be defined in the list of abbreviations.

(11)  Line 235:  "cytomegalovirus" (not "citomegalovirus").

(12)  Lines 216 and  236:  While on first sight it appears to be correct, the offiical definition of "MDS" is "myelodysplastic syndromes" (plural). See the latest WHO Classification:  Khoury et al., Leukemia 36: 1703-1719, 2022.

(13)  Lines 218 and 236:  It is now "Hodgkin lymphoma" (no longer "Hodgkin`s lymphoma").  Latest WHO Classification:  Alaggio et al., Leukemia 36: 1720-1748, 2022.

(14)  Line 240:  "Healthy donors were recruited as healthy donors"? Proposal: "Eighteen healthy donors were recruited as control population".

(15)  Lines 356 , 358, 372, 428:  The greek symbols in "γδ" of the T-cells with TCR cells does not show up in my preview of the manuscript. Elsewhere (for example line 182) the term "TCRγδ "is visible.

(16)  Lines 423-424:  The statement that multiple myeloma does not require "intensive chemotherapy" is in this form not correct. While there are nowadays also other options, chemotherapy is still the backbone of therapy.

Maybe the patients of this study did not receive chemotherapy in the setting of a university hospital. If so, then proposed rewording:  "... (mostly MM patients who in the present study did not receive intensive chemotherapy)".

Reviewer 4 Report

The article by Rodriguez-Mora et al. is well written and might be advisable for publication, although there are several important points in this research that should be clarified. For example, the AGE of the recipient with respect to the hematopoietic transplant, as well as the vaccination response are key and here they are considered lightly, grouping age groups biasedly. Thus, it is widely known that the older the age, the worse the vaccination response and the worse the severity of covid19. There are articles you should cite with this fact:

Age-dependent association of clonal hematopoiesis with COVID-19 mortality in patients over 60 years. Del Pozo-Valero M, et al. Geroscience. 2023 Feb;45(1):543-553.

Activating Killer-Cell Immunoglobulin-Like Receptors Are Associated With the Severity of Coronavirus Disease 2019. Bernal E, et al. J Infect Dis. 2021 Jul 15;224(2):229-240.

Table 1 shows that healthy people and patients are matched by age. It is not enough to say it, you have to show it. Another important point is that something obvious is given as a conclusion, the difference between an autotransplant and an allogeneic transplant is key, as far as an autotransplant behaves as a similar response to a normal individual. This is apparently normal. Also, when lymphocyte populations decrease, for example CD3+ in allogeneic transplantation and not in autologous transplantation, since immunosuppression is totally different in both types of transplantation.

Round 2

Reviewer 1 Report

The Change are acceptable. 

Reviewer 4 Report

The suggested changes have been done. This paper is advisable for publication.